# LoRA as an Implicit KL Regularizer in GRPO Fine-Tuning: From Theory to Practice

## Abstract

Low-Rank Adaptation (LoRA) is widely used for parameter-efficient reinforcement learning fine-tuning of large language models (LLMs), often together with an explicit KL penalty toward a reference policy. In this work, we analyze how the low-rank constraint itself can restrict parameter trajectories during gradient descent and limit the resulting policy shift. We study the learning dynamics of LoRA updates and derive an explicit upper bound on the KL divergence between the reference and updated policies that depends on the adapter rank. Empirically, in Group Relative Policy Optimization (GRPO) fine-tuning of several LLM families on reasoning benchmarks, we observe that removing the explicit KL penalty yields similar evaluation accuracy while reducing training time due to avoiding reference-policy evaluations. Our results provide theoretical grounding for KL-free fine-tuning with LoRA, maintaining reasoning performance while allowing for training speedups in practice.

## 1 Introduction

While large language models (LLMs) exhibit strong general abilities, their performance on multi-step reasoning remains limited (Patil & Jadon, 2025). Recent studies therefore explore reinforcement learning (RL) fine-tuning methods that optimize models via structured preferences rather than next-token prediction. Among these, Group Relative Preference Optimization (GRPO) extends Proximal Policy Optimization (PPO) (Schulman et al., 2017) and Direct Preference Optimization (DPO) (Rafailov et al., 2023) by learning from groupwise preferences over multiple model completions, and chains of thought (Shao et al., 2024).

Policy optimization approaches typically regularize the updated policy toward a reference using a Kullback-Leibler (KL) penalty (Schulman et al., 2015) to stabilize training with favorable error-averaging properties and linear horizon bounds (Vieillard et al., 2020), at the cost of increased computations due to multiple policy evaluations (Stiennon et al., 2020). Hu et al. (2022) introduced Low-Rank Adaptation (LoRA) which has been incorporated into RL pipelines to reduce memory and computational overhead (Wang et al., 2025), yet most existing implementations still retain the expensive KL regularization computation (Santacroce et al., 2023; Li et al., 2025). Recent empirical results suggest that removal of explicit KL penalties might not significantly alter final task performance during LoRA training (Yu et al., 2025; Liu et al., 2025; Li et al., 2025; Sun et al., 2023). Several recent works report empirical observations suggesting that removing explicit KL penalties during LoRA training may not substantially affect final task performance (Yu et al., 2025; Liu et al., 2025; Li et al., 2025; Sun et al., 2023). However, a principled understanding of the mechanisms driving these KL-free methods has not been formally investigated, and controlled empirical comparisons are largely absent. However, a principled understanding of the mechanisms driving these KL-free methods – and controlled empirical comparisons isolating this effect – remain largely absent.

We analyze the stability of KL-free LoRA training, develop a theoretical perspective on how the low-rank parameterization constrains policy updates, and provide empirical evidence supporting this mechanism. We summarize our contributions as follows. First, we derive an explicit upper bound on the KL divergence induced by LoRA updates, showing that it scales with the adapter rank. Specifically, we prove that the KL divergence between policies is bounded by a function that grows with the LoRA rank $r$ (Theorem 1). This bound formally establishes that low-rank constraints can limit policy drift during training. Secondly,

we show empirically that policy divergence scales with the LoRA rank and run large-scale GRPO fine-tuning controlled experiments on Gemma, Llama, and Qwen models. These experiments show that low-rank constraints provide sufficient stability, unlike full fine-tuning, where KL-free LoRA training preserves reasoning performance with reduced average training time.

## 2    Related Work

Parameter-efficient fine-tuning methods have become standard for adapting LLMs, offering significant reductions in memory and compute while maintaining competitive performance. Among these, LoRA (Hu et al., 2022) is particularly popular: it constrains weight updates to a low-rank subspace via a reparameterization. Instead of updating the full weight matrix $\mathbf{W_0} \in \mathbb{R}^{d \times k}$, it injects trainable matrices $\mathbf{A}$ and $\mathbf{B}$ such that $\Delta \mathbf{W} = \mathbf{BA}$, with $\mathbf{B} \in \mathbb{R}^{d \times r}$, $\mathbf{A} \in \mathbb{R}^{r \times k}$, and $r \ll \min(d, k)$.

Recent work has explored reinforcement-style fine-tuning to enhance the reasoning capabilities of LLMs. PPO, which requires a separate value model, is prone to instability and requires carefully tuned regularization (Schulman et al., 2017). Earlier approaches relied on KL penalties to constrain policy deviation from a reference model, stabilizing training but increasing computational cost (Schulman et al., 2015; Stiennon et al., 2020). Recent extensions include GRPO (Shao et al., 2024), which learns from groupwise preference signals. Several works have targeted computational overheads in GRPO. Yu et al. (2025); Liu et al. (2025) heuristically relax or remove the KL term in specific settings, though without theoretical justification or empirical evidence in general cases. Additional work has targeted the computational overhead of sampling multiple completions per prompt: Lin et al. (2025) propose pruning low-advantage completions, which dynamically reallocates GPU capacity, achieving substantial training speedups while preserving accuracy.

LoRA is widely adopted in RL fine-tuning of LLMs to reduce memory and computational costs (e.g., (Wang et al., 2025; Sun et al., 2023; Santacroce et al., 2023; Li et al., 2025)). Schulman & Lab (2025) demonstrate across policy gradient experiments that LoRA matches full fine-tuning even at rank 1, arguing on information-theoretic grounds that RL updates require very low representational capacity. They ablate LoRA rank and its effect on reward, but never consider LoRA as an implicit regularizer or compare it to other regularization methods. Wang et al. (2025) additionally shows that, even with extremely small LoRA rank, LoRA-based RL follows the reward dynamics of full-parameter GRPO with KL regularization across multiple architectures. Sun et al. (2023) empirically observe stability without KL regularization in a PPO setting in Llama experiments, but provided no theoretical explanation, nor any systematic study of LoRA's regularization effect and the impact of rank. Li et al. (2025) provide limited empirical evidence in restricted settings of fine-tuning KL-free GRPO under LoRA for computational reasons, but do not analyze how the LoRA rank influences policy divergence nor provide a theoretical characterization of this effect or systematic experimental comparisons.

GRPO extends PPO by learning from groupwise preferences over multiple model completions. For each prompt, GRPO samples a group of $G$ outputs and computes advantages relative to the group mean, eliminating the need for a separate value network. The objective is:

$$\mathcal{L}_{\text{GRPO}} = -\mathbb{E}_{\mathbf{x}, \{\mathbf{y}_g\}_{g=1}^G \sim \boldsymbol{\pi}_{\theta_{\text{old}}}} \left[ \frac{1}{G} \sum_{g=1}^G \min \left( \frac{\boldsymbol{\pi}_\theta(\mathbf{y}_g|\mathbf{x})}{\boldsymbol{\pi}_{\theta_{\text{old}}}(\mathbf{y}_g|\mathbf{x})} \hat{A}_g, \text{clip}\left( \frac{\boldsymbol{\pi}_\theta(\mathbf{y}_g|\mathbf{x})}{\boldsymbol{\pi}_{\theta_{\text{old}}}(\mathbf{y}_g|\mathbf{x})}, 1-\epsilon, 1+\epsilon \right) \hat{A}_g \right) - \beta D_{\text{KL}}(\boldsymbol{\pi}_\theta \| \boldsymbol{\pi}_{\text{ref}}) \right],$$

where $\hat{A}_g = r(\mathbf{x}, \mathbf{y}_g) - \frac{1}{G} \sum_{g'=1}^G r(\mathbf{x}, \mathbf{y}_{g'})$ is the advantage computed from reward $r(\cdot)$, $\epsilon$ is the clipping parameter, and $\beta$ controls KL regularization toward a reference policy $\boldsymbol{\pi}_{\text{ref}}$.

## 3    Problem Setting

We now analyze the effect of low-rank updates on model behavior in LoRA adaptation of Transformer-based language models. We model the update dynamics using single-layer neural networks, since LoRA is typically applied to linear projection layers in this setting (e.g., queries, keys, values, and output representations). We consider binary classification with a sigmoid activation and binary cross-entropy (BCE) loss. We analyze the effect of Stochastic Gradient Descent (SGD) updates on LoRA parameters by considering randomly sampled

Table 1: Summary of notation.

| Symbol | Dimension | Description |
|---|---|---|
| $\mathbf{W}_0$ | $\mathbb{R}^{d \times k}$ | Frozen pretrained weight matrix |
| $\mathbf{B}_i$ | $\mathbb{R}^{d \times r}$ | LoRA matrix at policy update step $i$ |
| $\mathbf{A}_i$ | $\mathbb{R}^{r \times k}$ | LoRA matrix at policy update step $i$ |
| $\mathbf{z}_i(\mathbf{x})$ | $\mathbb{R}^d$ | Logits: $\mathbf{z}_i(\mathbf{x}) = (\mathbf{W}_0 + \mathbf{B}_i \mathbf{A}_i)\mathbf{x}$ |
| $\boldsymbol{\pi}_{\theta_i}(\mathbf{x})$ | $\mathbb{R}^d$ | policy: $\boldsymbol{\pi}_{\theta_i}(\mathbf{x}) = \sigma(\mathbf{W}_0\mathbf{x} + \mathbf{B}_i \mathbf{A}_i \mathbf{x})$ |
| $\mathbf{y}_i$ | $\mathbb{R}^d$ | Ground-truth binary labels |
| $\mathbf{g}_i$ | $\mathbb{R}^d$ | $\mathbf{g}_i := \nabla_{\mathbf{z}_i(\mathbf{x})} \mathcal{L}_{\mathrm{BCE}}(\boldsymbol{\pi}_{\theta_i}(\mathbf{x}), \mathbf{y}_i)$ |
| $\mathbf{x}_e$ | $\mathbb{R}^k$ | Evaluation sample |
| $\mathbf{x}_i$ | $\mathbb{R}^k$ | Training sample at iteration |
| $i$ | – | Index of the current SGD update step |
| $t$ | – | Index of the final SGD update step |

training pairs $(\mathbf{x}_i, \mathbf{y}_i)_{i=1}^t$. We initialize the model with parameters $(\mathbf{A}_0, \mathbf{B}_0)$ and study how gradient steps influence the model's prediction on an evaluation input. The policy is parameterized by weights $\boldsymbol{\theta}_i$ at step $i$:

$$\boldsymbol{\pi}_{\theta_i}(\mathbf{x}_i) = \sigma\big(\mathbf{z}_i(\mathbf{x}_i)\big), \qquad \mathbf{z}_i(\mathbf{x}_i) := \mathbf{W}_0\mathbf{x}_i + \mathbf{B}_i\mathbf{A}_i\mathbf{x}_i,$$

where $\mathbf{x}_i \in \mathbb{R}^k$ is the input, $\mathbf{W}_0$ the fixed base weight, $\mathbf{A}_i, \mathbf{B}_i$ the trainable LoRA parameters, $\mathbf{z}_i(\mathbf{x}_i)$ the logits at step $i$ and $\sigma(\cdot)$ the sigmoid function.

In our binary classification problem, we define BCE loss to minimize:

$$\mathcal{L}_{\mathrm{BCE}}\left(\boldsymbol{\pi}_{\theta_i}(\mathbf{x}_i), \mathbf{y}_i\right) := -\mathbf{y}_i^\top \log\left(\boldsymbol{\pi}_{\theta_i}(\mathbf{x}_i)\right) - (\mathbf{1} - \mathbf{y}_i)^\top \log\left(\mathbf{1} - \boldsymbol{\pi}_{\theta_i}(\mathbf{x}_i)\right).$$

The next section considers the learning dynamics of LoRA (Ren & Sutherland, 2025) to characterize the influence on the policy. Specifically, we study the variation in the policy $\boldsymbol{\pi}_{\theta_0}(\mathbf{x}_e)$ induced by SGD steps. We ask the fundamental question:

> *How does this update influence the model's behavior, as measured by the change in the policy*
> $\Delta \log \boldsymbol{\pi}_{\theta_i}(\mathbf{x}_e) = \log \boldsymbol{\pi}_{\theta_i}(\mathbf{x}_e) - \log \boldsymbol{\pi}_{\theta_0}(\mathbf{x}_e)$ *for an arbitrary input* $\mathbf{x}_e$?

In the next section, we derive an explicit expression for this shift and characterize its structure and magnitude.

## 4 Theoretical Analysis of LoRA Updates on the KL Divergence

We start by considering the SGD updates with learning rate $\eta$ on the model parameters. Since the base weights $\mathbf{W}_0$ are frozen, we have $\nabla_{\mathbf{W}_0} \mathcal{L}_{\mathrm{BCE}} = 0$. We derive the gradients of the binary cross-entropy loss with respect to $\mathbf{B}_i$ and $\mathbf{A}_i$, which parameterize the low-rank update. The accumulated differences in parameter weights after $t$ SGD steps are denoted as:

$$\begin{aligned}
\mathbf{\Delta A}_t = \mathbf{A}_t - \mathbf{A}_0 &= -\eta \sum_{i=0}^{t-1} \nabla_{[\mathbf{A}_i]_{l,m}} \mathcal{L}_{\mathrm{BCE}}(\boldsymbol{\pi}_{\theta_i}(\mathbf{x}_i), \mathbf{y}_i) \\
&= -\eta \sum_{i=0}^{t-1} \mathrm{tr}\Big[\Big[\frac{\partial \mathcal{L}_{\mathrm{BCE}}}{\partial \mathbf{z}_i(\mathbf{x}_i)}\Big]^\top \frac{\partial \mathbf{z}_i(\mathbf{x}_i)}{\partial [\mathbf{A}_i]_{l,m}}\Big] \\
&= -\eta \sum_{i=0}^{t-1} \mathrm{tr}\Big[\mathbf{g}_i^\top \mathbf{B}_i \frac{\partial \mathbf{A}_i}{\partial [\mathbf{A}_i]_{l,m}} \mathbf{x}_i\Big] \\
&= -\eta \sum_{i=0}^{t-1} \mathrm{tr}\big[\mathbf{e}_l^\top \mathbf{B}_i^\top \mathbf{g}_i \mathbf{x}_i^\top \mathbf{e}_m\big] = -\eta \sum_{i=0}^{t-1} \big[\mathbf{B}_i^\top \mathbf{g}_i \mathbf{x}_i^\top\big]_{l,m},
\end{aligned} \tag{1a}$$

$$\Delta \mathbf{B}_t = \mathbf{B}_t - \mathbf{B}_0 = -\eta \sum_{i=0}^{t-1} \nabla_{[\mathbf{B}_i]_{l,m}} \mathcal{L}_{\text{BCE}}(\boldsymbol{\pi}_{\theta_i}(\mathbf{x}_i), \mathbf{y}_i)$$

$$= -\eta \sum_{i=0}^{t-1} \text{tr}\Big[\Big[\frac{\partial \mathcal{L}_{\text{BCE}}}{\partial \mathbf{z}_i(\mathbf{x}_i)}\Big]^\top \frac{\partial \mathbf{z}_i(\mathbf{x}_i)}{\partial [\mathbf{B}_i]_{l,m}}\Big]$$

$$= -\eta \sum_{i=0}^{t-1} \text{tr}\Big[\mathbf{g}_i^\top \frac{\partial \mathbf{B}_i}{\partial [\mathbf{B}_i]_{l,m}} \mathbf{A}_i \mathbf{x}_i\Big]$$

$$= -\eta \sum_{i=0}^{t-1} \text{tr}\big[\mathbf{e}_l^\top \mathbf{g}_i \mathbf{x}_i^\top \mathbf{A}_i^\top \mathbf{e}_m\big] = -\eta \sum_{i=0}^{t-1} \big[\mathbf{g}_i \mathbf{x}_i^\top \mathbf{A}_i^\top\big]_{l,m}, \tag{1b}$$

where $\mathbf{e}_m$ is the $m$-th basis vector where the $m$-th entry is 1 and all other entries are 0.

We study the perturbation induced by SGD updates over $t$ steps. After the update, if we evaluate back on $\mathbf{x}_e$ we can consider that the change on the logits to be perturbations induced by the optimizer step on $\mathbf{A}_0, \mathbf{B}_0$. The perturbation on the logit is due to the perturbation on the LoRA weights as follows:

$$\mathbf{z}_t(\mathbf{x}_e) = \mathbf{W}_0 \mathbf{x}_e + (\mathbf{B}_0 + \Delta \mathbf{B}_t)(\mathbf{A}_0 + \Delta \mathbf{A}_t)\mathbf{x}_e$$

$$= \underbrace{(\mathbf{W}_0 + \mathbf{B}_0 \mathbf{A}_0)\mathbf{x}_e}_{\mathbf{z}_0(\mathbf{x}_e)} + \Delta \mathbf{z}_t(\mathbf{x}_e), \tag{2}$$

where the update term can be further expressed with (Eq. (1a)) and (Eq. (1b)) as:

$$\Delta \mathbf{z}_t(\mathbf{x}_e) = -\eta \sum_{i=0}^{t-1} \Big(\mathbf{B}_0 \mathbf{B}_i^\top \mathbf{g}_i \mathbf{x}_i^\top \mathbf{x}_e + \mathbf{g}_i \mathbf{x}_i^\top \mathbf{A}_i^\top \mathbf{A}_0 \mathbf{x}_e - \eta \sum_{j=0}^{t-1} \mathbf{g}_i \mathbf{x}_i^\top \mathbf{A}_i^\top \mathbf{B}_j^\top \mathbf{g}_j \mathbf{x}_j^\top \mathbf{x}_e\Big).$$

To relate changes in parameter space to shifts in the output distribution, we begin by establishing a connection between the policy difference and the corresponding change in logits. We write the log-ratio shift vector $\Delta \log \boldsymbol{\pi}_{\theta_i}(\mathbf{x}_e) \in \mathbb{R}^d$ in coordinate-wise update as follows:

$$\Delta \log \boldsymbol{\pi}_{\theta_t}(\mathbf{x}_e) := \log \boldsymbol{\pi}_{\theta_t}(\mathbf{x}_e) - \log \boldsymbol{\pi}_{\theta_0}(\mathbf{x}_e), \tag{3}$$

where $\boldsymbol{\pi}_{\theta_t}(\mathbf{x}_e)$ is the updated policy and $\boldsymbol{\pi}_{\theta_0}(\mathbf{x}_e)$ is the reference policy. Then, we obtain the following first-order characterizations.

**Proposition 1** (Policy Variation from Parameters). *Let $t$ SGD updates induce a change in the model parameters from $(\mathbf{A}_0, \mathbf{B}_0)$ to $(\mathbf{A}_t, \mathbf{B}_t)$. Then, under a first-order approximation at the reference policy, with constant $\eta$, the change in the policy logits at $\mathbf{x}_e$ is:*

$$\Delta \log \boldsymbol{\pi}_{\theta_t}(\mathbf{x}_e) = \text{diag}(\mathbf{1} - \boldsymbol{\pi}_{\theta_0}(\mathbf{x}_e))\Delta \mathbf{z}_t(\mathbf{x}_e) + \mathcal{O}(\eta^2). \tag{4}$$

*Proof.* Substituting Eq. (2) for $\mathbf{z}_t(\mathbf{x}_e)$ and applying a first-order Taylor expansion of $\log \sigma(\mathbf{z}_t(\mathbf{x}_e))$ around the reference logits $\mathbf{z}_0(\mathbf{x}_e)$ results in

$$\Delta \log \boldsymbol{\pi}_{\theta_t}(\mathbf{x}_e) = \log \sigma\big(\mathbf{z}_0(\mathbf{x}_e) + \Delta \mathbf{z}_t(\mathbf{x}_e)\big) - \log \sigma\big(\mathbf{z}_0(\mathbf{x}_e)\big)$$

$$= \text{diag}\Big(\tfrac{d}{d\mathbf{z}} \log \sigma(\mathbf{z})\big|_{z=\mathbf{z}_0(\mathbf{x}_e)}\Big)\Delta \mathbf{z}_t(\mathbf{x}_e) + \mathcal{O}(\|\Delta \mathbf{z}_t\|^2)$$

$$= \text{diag}\big(\mathbf{1} - \boldsymbol{\pi}_{\theta_0}(\mathbf{x}_e)\big)\Delta \mathbf{z}_t(\mathbf{x}_e) + \mathcal{O}(\eta^2).$$

□

We now derive an expression for the KL divergence between $\boldsymbol{\pi}_t(\mathbf{x}_e)$ and $\boldsymbol{\pi}_0(\mathbf{x}_e)$ using only the log-ratio shift vector $\Delta \log \boldsymbol{\pi}_{\theta_i}(\mathbf{x}_e)$.

**Proposition 2** (Exponential Perturbation)**.** *Expressing the KL divergence in terms of the perturbations and the reference policy,*

$$D_{\mathrm{KL}}\big(\boldsymbol{\pi}_{\theta_t}(\mathbf{x}_e) \,\|\, \boldsymbol{\pi}_{\theta_0}(\mathbf{x}_e)\big) = \mathbb{E}_{j\sim\boldsymbol{\pi}_{\theta_0}(\mathbf{x}_e)}\left[e^{\Delta\log\boldsymbol{\pi}_{\theta_t,j}(\mathbf{x}_e)} \cdot \Delta\log\boldsymbol{\pi}_{\theta_t,j}(\mathbf{x}_e)\right].$$

*Proof.* Using Eq. (3), it follows that $\boldsymbol{\pi}_{\theta_t}(\mathbf{x}_e) = \boldsymbol{\pi}_{\theta_0}(\mathbf{x}_e) \cdot e^{\Delta\log\boldsymbol{\pi}_{\theta_t}(\mathbf{x}_e)}$. Substituting into the KL definition and taking expectation over $\boldsymbol{\pi}_{\theta_0}$ gives

$$D_{\mathrm{KL}}\big(\boldsymbol{\pi}_{\theta_t}(\mathbf{x}_e) \,\|\, \boldsymbol{\pi}_{\theta_0}(\mathbf{x}_e)\big) = \sum_{j=1}^{d} \boldsymbol{\pi}_{\theta_0} e^{\Delta\log\boldsymbol{\pi}_{\theta_t}(\mathbf{x}_e)} \Delta\log\boldsymbol{\pi}_{\theta_t}(\mathbf{x}_e)$$

$$= \mathbb{E}_{j\sim\boldsymbol{\pi}_{\theta_0}(\mathbf{x}_e)}\left[e^{\Delta\log\boldsymbol{\pi}_{\theta_t,j}(\mathbf{x}_e)} \cdot \Delta\log\boldsymbol{\pi}_{\theta_t,j}(\mathbf{x}_e)\right].$$

$\square$

We introduce our main result below, where $\lesssim$ denotes *with high probability.*

**Assumption 1.** *Let* $\mathbf{A}_0$, $\mathbf{B}_0$ *be random with independent, mean-zero, unit-variance, sub-Gaussian entries.*

**Assumption 2.** *Let* $\mathbf{x_i}$ *and* $\mathbf{x_e}$ *have independent, mean-zero, unit-variance, sub-Gaussian entries.*

**Assumption 3.** *Let the gradient norm be bounded by* $G$: $\|g_i\|_\infty \leq G$ *as is done in prior theoretical studies on the learning dynamics of LLMs (Ren & Sutherland, 2025).*

**Theorem 1** (KL Divergence Bound under Low-Rank Updates)**.** *The KL divergence between the reference* $\boldsymbol{\pi}_{\theta_0}(\mathbf{x}_e)$ *and updated* $\boldsymbol{\pi}_{\theta_t}(\mathbf{x}_e)$ *policies is bounded by:*

$$D_{\mathrm{KL}}\big(\boldsymbol{\pi}_{\theta_t}(\mathbf{x}_e) \,\|\, \boldsymbol{\pi}_{\theta_0}(\mathbf{x}_e)\big)$$

$$\leq \|\Delta\log\boldsymbol{\pi}_{\theta_t}(\mathbf{x}_e)\|_\infty \cdot e^{\left(\|\Delta\log\boldsymbol{\pi}_{\theta_t}(\mathbf{x}_e)\|_\infty\right)} + \mathcal{O}(\eta^2)$$

$$\leq \mu(r) \cdot e^{\mu(r)} \tag{5}$$

$$\lesssim C(2\sqrt{r} + \sqrt{d} + \sqrt{k})^2 e^{\left(C(2\sqrt{r}+\sqrt{d}+\sqrt{k})^2\right)} + \mathcal{O}(\eta^2) \tag{6}$$

$$\text{with } C := \eta\sqrt{k}Gt(1 + \eta\sqrt{k}G)^t \tag{7}$$

$$\text{and } \mu(r) := \sqrt{rk}\big(\|\mathbf{B}_0\|_2 \cdot \|\Delta\mathbf{A}_t\|_2 + \|\mathbf{A}_0\|_2 \cdot \|\Delta\mathbf{B}_t\|_2\big).$$

*Proof.* The KL divergence can be expressed as: $D_{\mathrm{KL}}\big(\boldsymbol{\pi}_{\theta_t}(\mathbf{x}_e) \,\|\, \boldsymbol{\pi}_{\theta_0}(\mathbf{x}_e)\big) = \mathbb{E}_{j\sim\boldsymbol{\pi}_{\theta_0}(\mathbf{x}_e)}\left[e^{\delta_j} \cdot \delta_j\right]$, where $\delta_j := \Delta\log\boldsymbol{\pi}_{\theta_t,j}(\mathbf{x}_e)$. Since $xe^x$ is increasing for all $x$, and $\boldsymbol{\pi}_{\theta_0}(\mathbf{x}_e)$ is a probability distribution, this yields the pointwise envelope:

$$D_{\mathrm{KL}}(\boldsymbol{\pi}_{\theta_t}\|\boldsymbol{\pi}_{\theta_0}) \leq \|\Delta\log\boldsymbol{\pi}_{\theta_t}(\mathbf{x}_e)\|_\infty \cdot e^{\left(\|\Delta\log\boldsymbol{\pi}_{\theta_t}(\mathbf{x}_e)\|_\infty\right)}. \tag{8}$$

Let $j$ index a row of the output, we now upper bound the individual coordinate $|\Delta z_t(\mathbf{x_e})_j|$, which contributes to the shift in $\Delta\log\boldsymbol{\pi}_{\theta_t}$.

$$|[\Delta\mathbf{z}_t(\mathbf{x}_e)]_j| = |\mathbf{b}_j^\top \Delta\mathbf{A}_t\mathbf{x}_e + \mathbf{e}_j^\top \Delta\mathbf{B}_t\mathbf{A}_0\mathbf{x}_e|$$

$$\leq |\mathbf{b}_j^\top \Delta\mathbf{A}_t\mathbf{x}_e| + |\mathbf{e}_j^\top \Delta\mathbf{B}_t\mathbf{A}_0\mathbf{x}_e|. \tag{9}$$

Here $\mathbf{b}_j^\top$ denotes the $j$-th row of $\mathbf{B}_0$, and $\mathbf{e}_j^\top$ is the standard basis row vector. We now bound each term separately. Each LoRA update matrix $\Delta\mathbf{A}_t$ and $\Delta\mathbf{B}_t$ has the form: $\Delta\mathbf{A}_t = -\eta\sum_{i=0}^{t-1}\mathbf{B}_i^\top\mathbf{g}_i\mathbf{x}_i^\top$, $\Delta\mathbf{B}_t = -\eta\sum_{i=0}^{t-1}\mathbf{g}_i\mathbf{x}_i^\top\mathbf{A}_i^\top$. Each term in the sum is an outer product, so $\Delta\mathbf{A}_t$ and $\Delta\mathbf{B}_t$ each have rank at most $r$ after $t$ steps. Using the inequalities: $\|\mathbf{M}\mathbf{v}\|_2 \leq \|\mathbf{M}\|_\mathrm{F} \cdot \|\mathbf{v}\|_2$, $\|\mathbf{M}\|_\mathrm{F} \leq \sqrt{r}\cdot\|\mathbf{M}\|_2 \rightarrow \|\mathbf{M}\mathbf{v}\|_2 \leq \sqrt{r}\cdot\|\mathbf{M}\|_2\cdot\|\mathbf{v}\|_2$ and under Assumption 2, $\|\mathbf{x_e}\|_2 \lesssim \sqrt{k}$ (Thm 3.1.1 in Vershynin (2018))

$$|\mathbf{e}_j^\top \Delta\mathbf{B}_t\mathbf{A}_0\mathbf{x}_e| \leq \|\Delta\mathbf{B}_t\mathbf{A}_0\mathbf{x}_e\|_2 \leq \sqrt{rk}\|\mathbf{A}_0\|_2\|\Delta\mathbf{B}_t\|_2, \tag{10a}$$

$$|\mathbf{b}_j^\top \Delta\mathbf{A}_t\mathbf{x}_e| \leq \|\mathbf{b}_j\Delta\mathbf{A}_t\mathbf{x}_e\|_2 \leq \sqrt{rk}\|\mathbf{B}_0\|_2\|\Delta\mathbf{A}_t\|_2. \tag{10b}$$

Define $\mu(r) := \sqrt{rk}\big(\|\mathbf{B}_0\|_2 \cdot \|\Delta\mathbf{A}_t\|_2 + \|\mathbf{A}_0\|_2 \cdot \|\Delta\mathbf{B}_t\|_2\big)$, from Eqs. (9), (10),then with Eqs. (4), (8):

$$D_{\mathrm{KL}}\big(\boldsymbol{\pi}_{\theta_t}(\mathbf{x}_e) \,\|\, \boldsymbol{\pi}_{\theta_0}(\mathbf{x}_e)\big) \leq \mu(r) \cdot e^{\mu(r)}.$$

This intermediate result depends on $r$, input size $k$ and implicitly, through the operator norm of the adapters, of the gradient bound $G$, the learning rate $\eta$, step count $t$, and of the output dimension $d$. We derive a bound independent of the operator norms and apply the reverse triangle inequality to Eq.(1) at step $i$, $\|\Delta\mathbf{A}_i\|_2 = \|\mathbf{A}_i - \mathbf{A}_0\|_2 \geq \big|\|\mathbf{A}_i\|_2 - \|\mathbf{A}_0\|_2\big|$:

$$\|\mathbf{A}_0\|_2 - \|\Delta\mathbf{A}_i\|_2 \leq \|\mathbf{A}_i\|_2 \leq \|\mathbf{A}_0\|_2 + \|\Delta\mathbf{A}_i\|_2. \tag{11}$$

Focusing on the upper bound of Eq. (11), and applying the same argument to $\|\mathbf{B}_i\|_2$:

$$\|\mathbf{A}_i\|_2 + \|\mathbf{B}_i\|_2 \leq \|\mathbf{A}_0\|_2 + \|\mathbf{B}_0\|_2 + \|\Delta\mathbf{A}_i\|_2 + \|\Delta\mathbf{B}_i\|_2.$$

Define $s_i := \|\mathbf{A}_i\|_2 + \|\mathbf{B}_i\|_2$, $s_0 := \|\mathbf{A}_0\|_2 + \|\mathbf{B}_0\|_2$ and add the inequalities of $\|\Delta\mathbf{A}_i\|_2$ and $\|\Delta\mathbf{B}_i\|_2$:

$$s_i \leq s_0 + \eta\sqrt{k}G\sum_{j=0}^{i-1} s_j, \quad i = 0, 1, \ldots, t-1.$$

This is precisely the form of the discrete Grönwall inequality (Clark, 1987). Applying the Grönwall lemma :

$$s_i \leq s_0 \prod_{j=0}^{i-1}(1 + \eta\sqrt{k}G) = s_0(1 + \eta\sqrt{k}G)^i.$$

Therefore, for $i < t$:

$$\|\mathbf{A}_i\|_2 + \|\mathbf{B}_i\|_2 \leq (\|\mathbf{A}_0\|_2 + \|\mathbf{B}_0\|_2) \cdot (1 + \eta\sqrt{k}G)^t.$$

Under Assumption 3, $\|\mathbf{g}_i\|_2 \leq G$, and under Assumption 2, $\|\mathbf{x}_i\|_2 \lesssim \sqrt{k}$.

$$\|\Delta\mathbf{A_t}\|_2 \leq \eta\sqrt{k}Gt\sum_{i=0}^{t-1}\|\mathbf{B}_i\|_2, \quad \|\Delta\mathbf{B_t}\|_2 \leq \eta\sqrt{k}Gt\sum_{i=0}^{t-1}\|\mathbf{A}_i\|_2.$$

We can further upperbound each LoRA weight:

$$\|\Delta\mathbf{A_t}\|_2, \|\Delta\mathbf{B_t}\|_2 \leq \eta\sqrt{k}Gt(\|\mathbf{A}_0\|_2 + \|\mathbf{B}_0\|_2) \cdot (1 + \eta\sqrt{k}G)^t.$$

Under Assumption 1, by Theorem 4.3.3 in Vershynin (2018):

$$\|\mathbf{A}_0\|_2 \lesssim \sqrt{r} + \sqrt{k}, \quad \|\mathbf{B}_0\|_2 \lesssim \sqrt{d} + \sqrt{r}.$$

$$\begin{aligned}
|[\Delta\mathbf{z}_t(\mathbf{x}_e)]_j| &\leq \sqrt{rk}\,(\|\mathbf{B}_0\|_2 \cdot \|\Delta\mathbf{A}_t\|_2 + \|\mathbf{A}_0\|_2 \cdot \|\Delta\mathbf{B}_t\|_2)\\
&\leq \eta\sqrt{k}Gt(1 + \eta\sqrt{k}G)^t \cdot (\|\mathbf{A}_0\|_2 + \|\mathbf{B}_0\|_2)^2\\
&\lesssim \eta\sqrt{k}Gt(1 + \eta\sqrt{k}G)^t \cdot (2\sqrt{r} + \sqrt{d} + \sqrt{k})^2.
\end{aligned}$$

And finally replacing into Eq. (8):

$$D_{\mathrm{KL}}(\boldsymbol{\pi}_{\theta_t}\|\boldsymbol{\pi}_{\theta_0}) \lesssim C(2\sqrt{r} + \sqrt{d} + \sqrt{k})^2 e^{\left(C(2\sqrt{r}+\sqrt{d}+\sqrt{k})^2\right)} + \mathcal{O}(\eta^2),$$

with $C := \eta\sqrt{k}Gt(1 + \eta\sqrt{k}G)^t$. $\qquad\square$

Theorem 1 follows from a first-order Taylor expansion around the reference policy, and thus provides a sharp characterization of the induced policy shift in the local regime where higher-order terms remain negligible. In this setting, the bound in Eq. (5) offers an interpretable link between LoRA rank and KL growth. Moreover, Eq. (6) isolates the dominant term through the factor $e^{C(2\sqrt{r}+\sqrt{d}+\sqrt{k})^2}$, making explicit how stability depends on the effective update magnitude. As the policy drifts further from the reference, higher-order corrections may contribute meaningfully, and the bound in Eq. (5) becomes less descriptive. Since $C := \eta\sqrt{k}Gt(1 + \eta\sqrt{k}G)^t$ in Eq. (7) depends only on known hyperparameters (learning rate $\eta$, step count $t$, and the clipping-controlled gradient bound $G$) the expression can be used to guide practical choices of $\eta$, $t$, and clipping so as to keep the exponential term controlled and ensure that rank remains a meaningful predictor of KL throughout training.

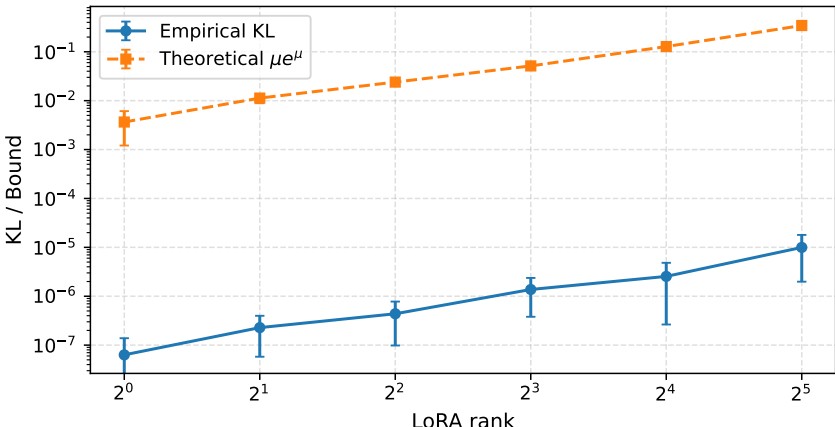

Figure 1: The empirical KL divergence and the KL bound (5) from Theorem 1. The error bars indicate the standard deviation across 20 seeds.

## 5    Experiments

We now examine the empirical implications of our theoretical analysis. We compare LoRA-based GRPO models trained with and without KL regularization, evaluating both policy divergence and task performance to assess LoRA's effectiveness as an implicit stabilizer. Our results suggest that LoRA can provide sufficient stability for reinforcement-style fine-tuning of LLMs without explicit KL penalties in practical scenarios.

### 5.1    Validation of the Theoretical KL Bound

We start with experiments to demonstrate the implications of Theorem 1 in a simplified setting consistent with its assumptions. We utilize the *Optdigits* dataset (Alpaydın & Kaynak, 1998) with a single-hidden-layer architecture with LoRA adapters ($d = 256, k = 64$). This design introduces a high-dimensional intermediate projection ($d \gg 1$) to prevent the rank collapse inherent to scalar binary outputs. This ensures the LoRA adapters operate within a subspace where $r \ll \min(d, k)$, satisfying the dimensionality assumptions of Theorem 1.

After pretraining, the base weights $\mathbf{W}_0$ are frozen, and the model is fine-tuned with LoRA for 30 epochs using a batch size of 64, a learning rate of $10^{-8}$, and gradient clipping at 0.001. For each rank, we compute both the empirical KL divergence between $\boldsymbol{\pi}_{\boldsymbol{\theta}_t}(\mathbf{x}_e)$ and $\boldsymbol{\pi}_{\boldsymbol{\theta}_0}(\mathbf{x}_e)$, and the theoretical upper bound (5) of Theorem 1. While these hyperparameters influence the magnitude of the gap between empirical values and the theoretical bound, they are sufficient for validating our theoretical framework in this controlled setting. Figure 1 shows that the empirical KL divergence scales upward with the LoRA rank, in line with the $\sqrt{r}$ dependence in (5). Theoretically, this scaling reflects the larger subspace accessible to the adaptation matrices at higher ranks, which allows for greater parameter variation and thus stronger policy deviation.

It is worth noting that the theoretical bound in Eq. (5) is loose by several orders of magnitude relative to the empirical KL, which is expected given the successive norm inequalities used in the proof of Theorem 1, each of which introduces slack. Despite this looseness, the bound remains valid across all 20 seeds, and the small variance across seeds confirms that the rank-dependent scaling is a robust property of the LoRA parameterization. This setup allows for precise tracking of the weight dynamics; in the next section, we scale to complex reasoning benchmarks in LLM training.

### 5.2    Scaling to LLMs: LoRA-Based GRPO with and without KL Penalty

To examine how the rank of LoRA adapters affects policy divergence, we conduct large-scale experiments across multiple LLM families on several reasoning tasks. For each rank setting, we also compare training

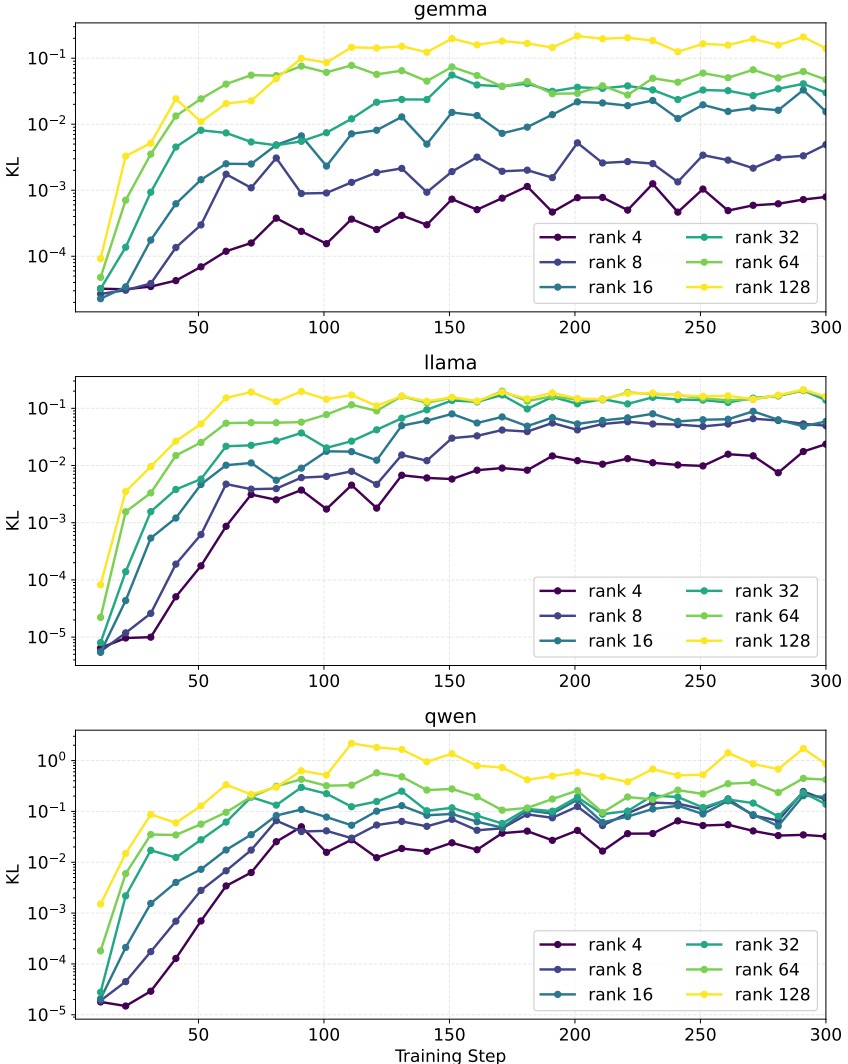

Figure 2: Mean KL divergence during GRPO training across LoRA ranks for Gemma, Llama, and Qwen. KL grows with rank. Higher ranks allow greater policy deviation.

runs with and without KL regularization in order to isolate its influence on accuracy and training dynamics. Models are trained with GRPO using AdamW (Loshchilov & Hutter, 2019) on the widely used GSM8K (Cobbe et al., 2021) dataset. We evaluate performance on the GSM8K evaluation split, GPQA (Rein et al., 2024), and MATH-500 (Hendrycks et al., 2021), which are established mathematical reasoning benchmarks (Shao et al., 2024).

Training hyperparameters are kept consistent across all model families. Preliminary experiments indicated that the standard learning rate of $5 \times 10^{-6}$ allowed all the models to reach a stable reward signal within 300 steps. The batch size is of 32 and the models were trained on a single Tesla V100 32GB GPU. We use $\beta = 0.05$ for the KL penalty, aligned with prior GRPO work (Shao et al., 2024).

To assess how the LoRA rank influences policy divergence and downstream reasoning performance, we perform GRPO fine-tuning across three LLM families: Gemma (et al, 2025), Llama (Grattafiori et al., 2024), and Qwen (Yang et al., 2025), using the GSM8K (Cobbe et al., 2021), GPQA (Rein et al., 2024), and MATH-500 (Hendrycks et al., 2021) benchmarks. Each model is trained with and without explicit KL regularization, holding all hyperparameters fixed to isolate the effect of LoRA rank.

Across all models, we observe a consistent growing trend of KL divergence with LoRA rank, as illustrated in Figure 2. The curves exhibit a near log-linear trend on a semilog scale: small ranks ($r \leq 16$) tightly

Table 2: Qwen2.5-3B-Instruct GRPO results with/without KL regularization across ranks. ΔAverage denotes the mean difference (w/o KL − with KL). It shows that removing KL reduces average runtime while maintaining accuracy (differences are within standard deviation).

| KL | $r$ | GPQA (↑) | GSM8K (↑) | MATH-500 (↑) | Runtime (min) (↓) |
|---|---|---|---|---|---|
| | 4 | **35.35** ± **3.41** | 75.97 ± 1.18 | 67.60 ± 2.10 | 139.17 |
| | 8 | 31.31 ± 3.30 | 75.59 ± 1.18 | 65.40 ± 2.13 | 119.64 |
| | 16 | 30.30 ± 3.27 | 77.48 ± 1.15 | 63.40 ± 2.16 | 109.61 |
| w/o | 32 | 30.81 ± 3.29 | 77.26 ± 1.15 | 66.20 ± 2.12 | 112.33 |
| | 64 | 34.85 ± 3.39 | **77.56** ± **1.15** | 66.60 ± 2.11 | 106.95 |
| | 128 | 32.83 ± 3.35 | **77.56** ± **1.15** | 65.00 ± 2.14 | 104.02 |
| | 4 | 34.85 ± 3.39 | 77.33 ± 1.15 | 64.80 ± 2.14 | 147.64 |
| | 8 | 33.84 ± 3.37 | 76.35 ± 1.17 | 67.20 ± 2.10 | 130.30 |
| | 16 | 25.25 ± 3.10 | 76.95 ± 1.16 | 65.60 ± 2.13 | 134.14 |
| w | 32 | 25.76 ± 3.12 | 77.26 ± 1.15 | 64.80 ± 2.14 | 125.40 |
| | 64 | 30.81 ± 3.29 | 76.27 ± 1.17 | **67.80** ± **2.09** | 112.79 |
| | 128 | 31.82 ± 3.32 | 77.48 ± 1.15 | 64.60 ± 2.14 | 87.01 |
| Δ *Average* | | +2.19 | -0.04 | -0.10 | -6.76 |

Table 3: Result on Gemma-3-1B-it.

| KL | $r$ | GPQA (↑) | GSM8K (↑) | MATH-500 (↑) | Runtime (min) (↓) |
|---|---|---|---|---|---|
| | 4 | 29.29 ± 3.24 | 48.22 ± 1.38 | 33.80 ± 2.12 | 342.85 |
| | 8 | 30.30 ± 3.27 | 48.52 ± 1.38 | 32.60 ± 2.10 | 328.48 |
| | 16 | 24.75 ± 3.07 | 49.13 ± 1.38 | 31.20 ± 2.07 | 321.71 |
| w/o | 32 | 24.24 ± 3.05 | 50.42 ± 1.38 | 31.40 ± 2.08 | 342.89 |
| | 64 | 24.24 ± 3.05 | 51.02 ± 1.38 | 31.20 ± 2.07 | 344.69 |
| | 128 | 27.27 ± 3.17 | **52.31** ± **1.38** | 32.60 ± 2.10 | 332.16 |
| | 4 | 23.74 ± 3.03 | 48.60 ± 1.38 | 31.40 ± 2.08 | 343.24 |
| | 8 | 24.75 ± 3.07 | 48.37 ± 1.38 | 31.80 ± 2.08 | 353.41 |
| | 16 | 25.25 ± 3.10 | 50.42 ± 1.38 | **34.20** ± **2.12** | 346.22 |
| w | 32 | **33.33** ± **3.36** | 48.82 ± 1.38 | 31.80 ± 2.08 | 364.34 |
| | 64 | 25.76 ± 3.12 | 48.07 ± 1.38 | 31.20 ± 2.07 | 366.42 |
| | 128 | 31.31 ± 3.30 | 46.40 ± 1.37 | 32.80 ± 2.10 | 350.97 |
| Δ *Average* | | -0.68 | +1.49 | -0.07 | -18.30 |

constrain the policy shift, whereas larger ranks ($r \geq 64$) produce an order-of-magnitude increase in KL. This pattern aligns with the theoretical insight that the effective stability region of the updates expands with the rank-dependent geometry of LoRA updates. The effect is most pronounced in Gemma and Llama, where KL grows steadily before plateauing, and is more variable in Qwen, which exhibits transient peaks at high ranks. Despite the increase in KL with rank, task performance remains stable across all configurations. As shown in Tables 2–4, accuracy differences between runs with and without KL regularization are minor and within the reported standard deviations. This suggests that LoRA provides sufficient regularization to maintain policy stability during GRPO fine-tuning without an explicit KL term, while also reducing computational cost on average. Across models, mean runtime decreases by roughly 7–20% without KL, reflecting both fewer forward passes (no reference-policy evaluation) and mainly faster convergence of the policy gradients. For small to mid-scale models, the KL penalty may be redundant because LoRA's low-rank parameterization already constrains policy updates, which is aligned with previous findings (Schulman & Lab, 2025) observing that LoRA matches full fine-tuning in RL even at minimal rank.

To disentangle the contributions of low-rank constraints and explicit KL regularization, we conduct a controlled comparison on Gemma-3-1B-it with four configurations: LoRA (rank 16) and full fine-tuning, each with and without KL regularization. Fig. 3a shows that the low-rank constraint is the dominant factor controlling KL magnitude. Both rank-16 runs remain an order of magnitude below their full fine-tuning counterparts throughout training, regardless of KL regularization. In contrast, the explicit KL penalty produces only a modest downward shift in both settings, where the LoRA stabilization can operate independently of explicit regularization at a *large scale*. Fig. 3b reveals a striking difference in how the two methods respond to removing KL regularization. In full fine-tuning, final rewards drop to zero without KL

Table 4: Results on Llama-3.2-3B-Instruct.

| KL | $r$ | GPQA ($\uparrow$) | GSM8K ($\uparrow$) | MATH-500 ($\uparrow$) | Runtime (min) ($\downarrow$) |
|---|---|---|---|---|---|
| w/o | 4 | $26.77 \pm 3.15$ | $68.23 \pm 1.28$ | $43.60 \pm 2.22$ | 176.26 |
| | 8 | $26.26 \pm 3.14$ | $68.08 \pm 1.28$ | $41.80 \pm 2.21$ | 172.80 |
| | 16 | $30.30 \pm 3.27$ | $68.23 \pm 1.28$ | $43.60 \pm 2.22$ | 161.34 |
| | 32 | $\mathbf{32.83} \pm \mathbf{3.35}$ | $68.99 \pm 1.27$ | $42.20 \pm 2.21$ | 151.89 |
| | 64 | $31.82 \pm 3.32$ | $\mathbf{69.83} \pm \mathbf{1.26}$ | $\mathbf{44.80} \pm \mathbf{2.23}$ | 159.62 |
| | 128 | $26.26 \pm 3.14$ | $68.01 \pm 1.28$ | $42.40 \pm 2.21$ | 152.48 |
| w | 4 | $27.27 \pm 3.17$ | $68.76 \pm 1.28$ | $42.00 \pm 2.21$ | 182.12 |
| | 8 | $30.30 \pm 3.27$ | $68.16 \pm 1.28$ | $43.60 \pm 2.22$ | 193.30 |
| | 16 | $27.27 \pm 3.17$ | $68.39 \pm 1.28$ | $41.80 \pm 2.21$ | 189.47 |
| | 32 | $28.28 \pm 3.21$ | $67.48 \pm 1.29$ | $43.60 \pm 2.22$ | 173.53 |
| | 64 | $26.77 \pm 3.15$ | $67.78 \pm 1.29$ | $\mathbf{44.80} \pm \mathbf{2.23}$ | 181.67 |
| | 128 | $29.29 \pm 3.24$ | $69.60 \pm 1.27$ | $43.60 \pm 2.22$ | 173.11 |
| $\Delta$ *Average* | | +0.84 | +0.20 | -0.17 | -20.13 |

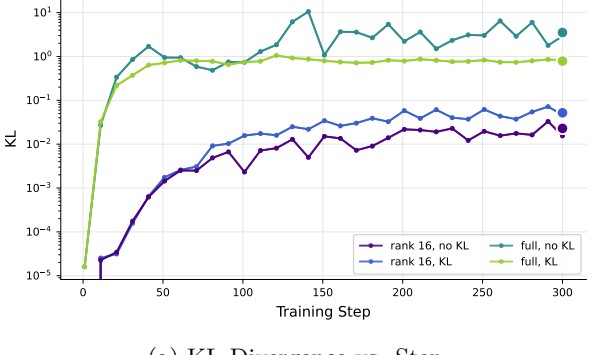

(a) KL Divergence vs. Step

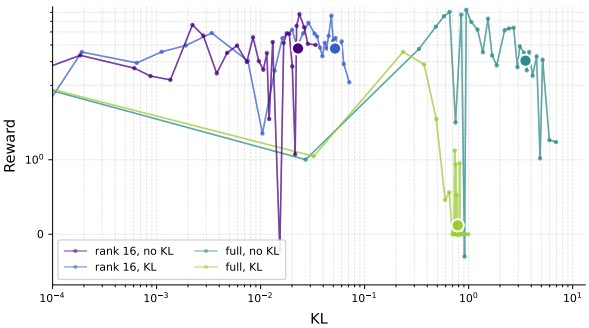

(b) Reward vs. KL Divergence

Figure 3: Ablation on the effect of LoRA and explicit KL regularization on policy divergence and performance under GRPO of Gemma-3-1B-it. (a) KL divergence over training steps, comparing the effect of low-rank adaptation (LoRA, rank 16) against explicit KL regularization on KL divergence magnitude. (b) Reward as a function of KL divergence. To characterize the relationship between policy shift and final performance, we report the average KL and reward over the last 50 training steps as a summary point for each configuration.

but remain at 3.0 with KL. However, LoRA rank 16 shows no such degradation: performance with and without KL regularization remains nearly identical at just below 4.0. This suggests that overall the low-rank constraint itself can provide regularization for fine-tuning.

Taken together, these results extend the controlled findings of Section 5.1 to the LLM scale: the LoRA rank directly governs the scale of policy divergence, and the theoretical $\sqrt{r}$ dependence holds qualitatively in full model training. The empirical stability of unregularized runs relates our main result that LoRA acts as an implicit policy stabilizer. Our comparison on Gemma-3-1B-it shows this mechanism operates independently: the low-rank constraint dominates KL control at a scale an order of magnitude larger than explicit regularization, and prevents the performance collapse observed in unregularized full fine-tuning. LoRA provides both a theoretical and practical alternative to explicit KL penalties in preference-based reinforcement learning for LLMs.

## 6 Conclusion

This work establishes a theoretical and empirical connection between the low-rank geometry of LoRA and the stability of policy optimization in reinforcement-style fine-tuning. We demonstrated that constraining parameter updates to a rank-$r$ subspace imposes an intrinsic upper bound on the KL divergence, thereby limiting policy drift as a function of the adapter rank. This result provides a principled explanation for

the empirical stability of LoRA-based reinforcement fine-tuning methods and offers a theoretical basis for KL-free training. In experiments across multiple LLM architectures, LoRA-based GRPO remains stable and achieves similar accuracy without the KL penalty. Beyond its parameter efficiency, LoRA therefore acts as an implicit stability mechanism on the policy updates, offering both theoretical grounding and practical gains for scalable, KL-free GRPO adaptation of LLMs.

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
