# OpenReview forum: "LoRA as an Implicit KL Regularizer in GRPO Fine-Tuning: From Theory to Practice"
_TMLR — Withdrawn by Authors_

### Review · Reviewer_MrNX · 2026-04-11

**Summary Of Contributions:**

The paper studies whether explicit KL regularization is still necessary in GRPO fine-tuning when the policy is adapted with LoRA. The main contribution is a theoretical argument that LoRA's low-rank constraint implicitly limits policy drift, together with a rank-dependent upper bound on KL divergence in a simplified single-layer setting. Empirically, the authors validate the rank-KL trend in a toy experiment and then compare GRPO with and without KL across Gemma, Llama, and Qwen models on GSM8K, GPQA, and MATH-500 for LoRA ranks from 4 to 128. The main finding is that KL tends to increase with rank, while downstream accuracy differences between KL and KL-free runs are usually small relative to the reported variation, suggesting that LoRA may provide useful implicit stabilization.

Strengths
- The paper addresses a practically relevant question, since removing explicit KL could reduce the cost of RL fine-tuning.
- The theory offers a clear qualitative link between LoRA rank and policy drift, which gives a plausible mechanism for the observed stability.
- The experiments are reasonably systematic in sweeping ranks across three model families and reporting both KL and task accuracy.
- The Gemma ablation comparing LoRA and full fine-tuning, each with and without KL, is helpful for isolating the role of the low-rank constraint.

Weaknesses
- The theoretical setting is far from the actual LLM training setup, so it mainly provides intuition rather than strong support for the practical claims.
- The KL bound is very loose and only justified in a local first-order regime.
- The empirical scope is narrow for the paper's conclusion: only 1B-3B models, one training dataset, and relatively short 300-step runs are considered.
- The results mostly show similar performance without KL rather than a strong, consistent advantage.
- Reproducibility details for the LLM experiments are unclear, including the number of seeds and what the reported standard deviations represent.

**Additional Comments:**

NA

**Audience:**

Yes

**Audience Explanation:**

The interaction between LoRA, KL regularization, and RL fine-tuning efficiency is relevant to researchers working on LLM alignment, preference optimization, and parameter-efficient training. Even though the evidence is not yet fully conclusive, the paper studies a concrete and timely question with practical implications for reducing training cost in GRPO-style methods.

**Claims And Evidence:**

No

**Claims Explanation:**

The paper provides suggestive but not fully convincing evidence. The empirical results do support the narrower claim that KL tends to grow with LoRA rank and that, in these particular small-to-mid-scale GRPO runs, removing KL often does not hurt final benchmark accuracy much. However, the main practical claim is broader than the evidence: the theory is derived in a simplified single-layer binary-classification setting with SGD, while the actual experiments are autoregressive transformer GRPO with AdamW, and the paper does not clearly justify how much of the theory transfers. In addition, the LLM evaluation is limited to 1B-3B models, one training dataset, and 300-step runs, so the evidence is not yet strong enough to conclude that explicit KL is generally unnecessary in LoRA-based GRPO.

**Requested Changes:**

Requested Changes
- Clarify the intended scope of the theory in Sec. 4. The paper should explicitly state which parts are only heuristic intuition for transformer GRPO and which parts are expected to transfer beyond the simplified sigmoid/BCE setting.
- Strengthen the empirical evidence for the main claim by adding at least one longer-horizon experiment or one larger model, so the conclusion is not limited to short 300-step runs on 1B-3B models.
- Report missing experimental details for the LLM study, including the number of seeds, what the reported standard deviations measure, and preferably confidence intervals or paired tests for with-KL versus without-KL comparisons.
- Clarify the runtime protocol in Sec. 5.2 and Tables 2-4. If all runs use a fixed 300-step budget, explain exactly how runtime was measured and why the savings vary substantially by rank.

Optional
- Add a stronger regularization comparison, such as tuned smaller-beta KL, adaptive-KL GRPO, or another lightweight baseline, to better support the claim that explicit KL is redundant rather than just weakly helpful at beta = 0.05.
- Repeat the LoRA-versus-full-fine-tuning ablation from Fig. 3 on at least one additional model family.
- Discuss more explicitly that the present evidence supports a narrower claim about small-to-mid-scale settings, unless additional experiments justify a broader conclusion.

---

### Review · Reviewer_gpi5 · 2026-04-14

**Summary Of Contributions:**

The paper analyzes the usage of LoRA in GRPO fine-tuning. The authors start by analyzing the optimization procedure theoretically to provide a bound on the KL divergence. Subsequently, they run experiments to verify that their bound holds, and run some experiments with LoRA with and without the KL penalty -- the claim being that sometimes the KL penalty is not necessary.

Strengths:
- The topic of the paper is quite important, as GRPO fine-tuning is a very popular technique. The insight that LoRA acts as a regularizer is interesting and worth further investigation.
- The derivation of the theorem seems correct although I didn't check the proofs line-by-line.
- The paper is well-written and the thought process is easy to follow.

Weaknesses:
- Ultimately the results presented here are not yet useful in practice. The bound presented by the authors, as they themselves admit is very loose. I am also not sufficiently convinced by the experiments -- the standard deviation is much higher than the differences of the means, making any conclusions difficult. Additionally, to have rigorous statements here, an extensive hyperparameter search would be required -- for now the authors set learning rate, KL coefficient, etc. to constant values, while in practice these values interact with each other in non-obvious ways.
- I don't think the theoretical analysis applies directly to the general LLM case, as the authors assume that the object of interest is a single-layer MLP. In practice, the LoRA layers are often applied to earlier blocks, and in effect the perturbation of the output policy can be significantly larger due to the composition of layers.
- I'd appreciate more details about how the KL was applied in the experiments, in particular there have been some papers recently that suggest that some "naive" applications of KL lead to incorrect gradients [1, 2, 3].

[1] Shah, V., Obando-Ceron, J., Jain, V., Bartoldson, B., Kailkhura, B., Mittal, S., ... & Courville, A. (2025). A Comedy of Estimators: On KL Regularization in RL Training of LLMs. arXiv preprint arXiv:2512.21852. \
[2] Tang, Y., & Munos, R. (2025). On a few pitfalls in kl divergence gradient estimation for rl. arXiv preprint arXiv:2506.09477. \
[3] Zhang, Y., Liu, Y., Yuan, H., Yuan, Y., Gu, Q., & Yao, A. C. C. (2025). On the design of kl-regularized policy gradient algorithms for llm reasoning. arXiv preprint arXiv:2505.17508.

**Audience:**

No

**Audience Explanation:**

I don't think the results presented here in the current form are sufficiently interesting to the TMLR audience. If it was possible to provide a tighter bound and more extensive empirical experiments, this study would be very interesting, but as it is now there is not enough to say anything significant.

**Claims And Evidence:**

No

**Claims Explanation:**

While I find the theoretical claims to be well-supported, the empirical part is more problematic. As mentioned in the text field above, the high noise in the results combined with lack of a bigger hyperparameter search makes it difficult to draw significant conclusions from these results.

**Requested Changes:**

- Make the experimental sweep significantly bigger, show that the results are statistically significant in more robustly analyzed settings.
- Get a tighter theoretical bound -- obviously, this is easy to say and much harder to do, but in the current form I'm doubtful whether this bound is useful.
- Add more details on the experimental setup, ideally including the code.

---

### Review · Reviewer_5CjZ · 2026-04-18

**Summary Of Contributions:**

This paper studies whether LoRA can serve as an implicit regularizer in GRPO fine-tuning, potentially reducing or even removing the need for an explicit KL penalty to a reference policy. The main theoretical claim is that the KL divergence between the updated policy and the reference policy is upper bounded by a quantity that increases with the LoRA rank. Empirically, the paper evaluates GRPO fine-tuning with and without explicit KL regularization across several LLM families and reasoning benchmarks, and reports that removing the KL term largely preserves evaluation accuracy while often reducing runtime.

**Audience:**

Yes

**Audience Explanation:**

Yes. The paper studies a practically relevant question in LLM post-training, whether LoRA can reduce the need for explicit KL regularization in GRPO, and both the theoretical perspective and empirical findings could be of interest to researchers working on parameter-efficient fine-tuning, RL for LLMs, and efficient reasoning model training.

**Claims And Evidence:**

No

**Claims Explanation:**

Partially. The empirical results support the claim that LoRA-based GRPO can often remain stable without an explicit KL term, and the observed rank–KL trend is consistent with the paper’s intuition. However, the theoretical evidence is less convincing, since it is derived in a highly simplified setting and the bound is quite loose. It is not obvious how strongly the resulting KL bound transfers to multi-layer Transformer policies trained with autoregressive objectives and token-level softmax distributions.

**Requested Changes:**

1. Discuss more explicitly the gap between the simplified theoretical setting and real Transformer-based LLM fine-tuning.
2. More the empirical validation will enhance this paper. For example with larger-scale models, longer training runs, or additional evidence on when explicit KL regularization does or does not matter.

---

### Note · Authors · 2026-04-30

I have read and agree with the venue's withdrawal policy on behalf of myself and my co-authors.